

# Open Access: Strengths, Weaknesses, Opportunities, and Threats.

## An Editorial

Geoffrey Bodenhausen

10

Ecole Normale Supérieure-PSL Research University, Département de Chimie, 24 rue Lhomond, F-75005 Paris, France.

15





**Abstract**

About a year after launching "Magnetic Resonance" it seems appropriate to publish a few editorial
5    remarks about the strengths, weaknesses, opportunities, and threats of Open Access, motivated by an
analysis of the Ethics Committee of the French CNRS and by a debate organized by the Royal Dutch
Academy of Science.

10    **Key words**

Open Access. Plan S. Opinion of the French CNRS Ethics Committee. Webinar on Open Access, the
Royal Dutch Academy of Science. Publishing business models.



At the beginning, in the first half of 2019, when we decided to launch *Magnetic Resonance*, our motivation appeared to be quite straightforward. Peer-reviewed publications are widely regarded as a cornerstone of science. Publishing should therefore remain protected from commercial interests, and

should remain clean and ethical, as has been the case since the XVII[th] century when the ethics of scientific publishing were first laid down. It was only in the mid-1990's that "Chemical Abstracts" were converted from clumsy volumes into on-line services that could be readily exploited to determine $h$-indices and other indicators of "prestige" and "popularity". As if scientists needed to be promoted like Bollywood's movie stars. As if the creativity of Albert Overhauser, Al Redfield or John Waugh could be measured in

terms of citations, like the glamour and talent of Liz Taylor, Grace Kelly or Gérard Depardieu can be measured in terms of box office revenue. It was also in the mid-1990's that some commercial companies (primarily RLX/Elsevier, Springer/Nature, Cell Press, and Wiley) seized the opportunity to extort outsized profits on the back of public research. They were soon followed by prestigious learned societies such as the American Chemical Society (ACS), the American Association of the Advancement of Science

(AAAS), the Royal Society of Chemistry (RSC), and a few others, who realized that they too could make a hefty fortune with little effort. Publishers and learned societies alike fueled the lethal fashion of bibliometrics, with the paradoxical support of both researchers and administrators, the former because bibliometrics comforted their narcissistic aspirations, the latter because bibliometrics simplified their evaluations. Publishers and learned societies vigorously defend the supposed virtues of impact factors. At

meetings of Editorial Boards, Elsevier's representatives report extensively on impact factors and downloads. More traditional metrics, such as the time it takes from submission to publication, have lost much of their relevance since the advent of repositories and Google Scholar alerts. We scientists know better than anybody that the number of citations of some "peak" papers published in such-and-such a journal is a pathetic measure of the *average* quality of papers published in the same journal. Even for

individual papers, the number of "downloads" is a poor measure of their novelty. Can one compare a popular piece of NMR software with an invention like Albert Overhauser's? Shouldn't scientists refuse to see their work degraded to mere merchandise? Shouldn't we refuse to measure the quality of our work

in terms of a single number, knowing that the value of cultural creation – by it in music, plastic art, or research - cannot be measured in a single dimension? Can anyone imagine assigning marks to paintings? Does anybody wish to rank Leonardo da Vinci, Francesco de Goya, or Paul Cézanne? One could make similar arguments *ad absurdum* for classical composers, contemporary jazz, etc.

Negotiations between publishers and libraries are often coordinated on a national level, for example by the French Couperin consortium or the German DEAL. When they come to agree on a compromise, the terms are invariably hidden from the public by "non-disclosure agreements". In our view, the secrecy of these negotiations confirms our suspicions. Indeed, the takeover of a noble enterprise by multinational

profit seekers is best hidden from the public eye.

The Ethics Committee (COMETS) [1] of the French CNRS has published a remarkable in-depth 'opinion' (*avis,* perhaps better translated as *white paper*) that contains a wealth of useful information, such as definitions of Diamond, Gold, Green and Hybrid models, Transformative agreements, Plan S, DORA,

Article Processing Charges (APCs), archives like BioRxiv and HAL, pirate websites such as 'Sci-hub', undue profits raked up by publishers, predatory journals, negotiating agreements with publishers, etc. "The excessive profits of the major publishers encourage researchers to circumvent intellectual property rights with a clear conscience, while these same publishers, even if they file a complaint, can ultimately only turn a blind eye to these breaches of the law which, after all, disseminate their output."


Open access to scientific publications opens many stimulating new perspectives. The CNRS 'opinion' not only describes different modalities but also examines some of their possibly perverse consequences. Indeed, if more and more open access journals have adopted peer review, they must be distinguished by their way of recovering the costs of publication. Most of the time, open access journals require the

payment of "APC" (Article Processing Charges) either by the authors or by the organizations on which they depend. The CNRS 'opinion' analyzes in detail the different modalities (Diamond, Gold, Green and Hybrid models). If we are not careful, these models may lead to unfair systems which not only create

inequities between researchers but which generate undue profits for publishers, thanks to public investment and the work of scientists who not only produce the research but also ensure its evaluation, free of charge. Another perverse effect is the multiplication of editorial offers for reduced APC rates, without any guarantee of scientific rigor. Such offers artificially multiply the number of publications,

some of which may be qualified as questionable or even fraudulent. The CNRS 'opinion' provides information on procedures that allow the free deposit of research documents on open access platforms and their immediate access to all. The 'opinion' explains why we should abide by DORA principles and adopt *Creative Commons* licenses. It recommends strengthening the interoperability of international open archives. The CNRS 'opinion' also shows how researchers can deposit preprints of their articles on so-

called pre-print servers even before their evaluation, thus communicating them without delay to the entire community, which offers an opportunity to discuss and improve them. If the work is not subject to any evaluation, one can nevertheless organize reviewing through *Peer Community In* (PCI).[2] Genuine scientific forums can thus be created. The CNRS 'opinion' identifies many innovative and little-known forms of APC-free publication. For example *Epi-reviews* offer open access publication with an evaluation

by researchers without calling on private publishers. The *Open Edition platform* offers a complete electronic publishing infrastructure for human sciences without any APC payments, with free access to publications in HTML format. The CNRS 'opinion' not only offers a very complete inventory of scientific publications but also a detailed analysis of the advantages and disadvantages of each format. It provides figures on the profits of the publishers that deserve to be decried. Finally, it analyzes the consequences of

open publication on the evaluation of researchers.

The full text of the CNRS 'opinion' is attached as Appendix to this editorial in *Magnetic Resonance*.[3] It offers an up-to-date, virtually encyclopedic source of valuable information, complete with an exhaustive glossary. One comes to realize that the crisis of scientific publishing has reached amazing proportions.

Thus, in 2017, more than 1,000 publishers, many of whom should potentially be regarded as "predatory", have emerged and marketed approximately 10,000 journals!

**MAGNETIC RESONANCE**
Discussions

Since 2018, European funding agencies have applied pressure towards Open Access with a scheme called Plan S. Many of us have a hard time following the pros and cons of Plan S. In a nutshell: for a journal to be compliant with Plan S, there should be no pay-walls, no embargo periods, and no hybrid deals. Author page charges (APCs) should not be paid by authors, but by funders or governments. All papers (including

of course, those submitted to Science, Lancet, Cell or Nature) must be made instantly available in a public repository. Copyright should remain in the hands of the authors. Transformative Model Agreements are regarded with suspicion. Non-compliance can be sanctioned: in the Netherlands, the Dutch Research Council (NWO) plans to withhold 2.5 % of grants if the authors do not comply.

In accordance with its policy, the views of the Ethics Committee of the CNRS are as fair and balanced as they could be. We found it stimulating to compare these politically unassailable views with some controversial commentaries. In this spirit, we recommend a Webinar on Open Access [3] organized by the Royal Dutch Academy of Science (KNAW) on June 23rd 2020. Some believe that the Netherlands are close to realizing full OA and optimistically expect this fashion soon to sweep across the entire globe.

Others fear that Plan S may turn out to be divisive, since it is supported neither by Germany (where the authorities tend to favor the transformative route) nor by China, nor by the USA, who favor their own journals, provide little funding for Gold or Hybrid OA. The USA tend to rely on repositories such as PubMed, arXiv, ChemRxiv, BioRxiv and HAL, that massively archive papers of biomedical, physical and chemical interest. The Webinar on Open Access [4] mentions that the effects of Plan S on poor

countries and charitable foundations are unpredictable, as are the risk of increasing prices, and the possible loss of freedom to choose journals based on the reputation of their Editorial Boards. Not all OA journals match the standards of well-known journals. Hefty author page charges (APCs) such as those practiced by *Scientific Reports* (Nature) or *Science Advances* (AAAS) worsen inequalities between institutions and authors. Indeed, "gold open access" (i.e., where the author or his employer must pay) is not necessarily

an attractive route for human and social sciences, or, more generally, for retired faculty, whose institutions may be reluctant to support access to journals.

Against this complex background, we started "*Magnetic Resonance*" at the end of 2019. We were glad to make a deal with Copernicus Publications, a not-for-profit publishing company that specializes on producing quality Open Access journals. We put together a wonderful Editorial Board. Then came a series of wake-up calls.

*Magnetic Resonance* has now been in existence for about a year. Most papers received so far come from European laboratories, since the USA seem to ignore our initiative, despite our efforts to appoint board members who work in the USA. Most papers come from physics, few from chemistry, and hardly any from biology. We attribute this to the sad fact that the biomolecular community (NMR, XRD or CryoEM alike) appears to cultivate an unbridled passion for impact factors. The vicious role of these metrics appears to be less perverse in physics and chemistry than in biology. Some major journals like PNAS and Nature clearly favor biology – possibly because there are so many wealthy biomolecular NMR laboratories that their mutual citations are bound to flourish. In our jaded assessment, their creativity is not always on a par with the founding fathers of magnetic resonance, save a few breakthroughs, such as the pursuit of minor conformations in proteins and nucleic acids. When I watch a myriad of biomolecular structures, I like to think of Ernest Rutherford, who may (or may not) have said that "there are physicists and butterfly collectors".

Some well-off colleagues told us that the traditional system actually offers good value for money. True, publications cost far less than salaries and sophisticated instrumentation. Yet in 2014, the first year of my "advanced" ERC grant, I spent about € 60,000 on "hybrid" OA fees, the equivalent of a full post-doctoral salary. Money that could have been spent better. Not to mention that our libraries in France and Switzerland spent millions on subscriptions, and that many authors frequently sacrifice their time as referees, unpaid as always.

25

Some respectable colleagues came to the rescue of learned societies such as the American Chemical Society (ACS) and the Royal Society of Chemistry (RSC). I fail to understand their loyalty. Personally,

in the course of my 35-year career, I have paid a fortune in membership fees to the Swiss, American and French chemical societies, but I do not remember ever getting any significant benefits in exchange, except some reduced conference fees that however continue to generate comfortable revenues. (I would make an exception for the Groupement AMPERE, the owner of *Magnetic Resonance*, who signed an agreement

with Copernicus Publications to produce our journal and also acts as a sort of insurance company for EUROMAR and other meetings.)

While *Magnetic Resonance* struggles for recognition, some of the profit-seeking companies and societies mentioned above attempt to defend their market share. Thus, the historical *Journal of Magnetic*

*Resonance* is now backed up by the new *Journal of Magnetic Resonance Open*. It is not yet clear that this initiative will be successful. Elsevier and other profitable publishers will probably be forced to make further concessions. It is noteworthy that they remain unbowed in many key negotiations, with the German DEAL, the University of California, Harvard, etc. When Cell and Lancet will soon become compliant, it will be interesting to see if their profits shrink from a hefty 35-40% to a more reasonable 5-

10%. Some may simply abandon ship – if major airlines, airplane and car manufacturers are facing bankruptcy in the wake of the current economic crisis, why should taxpayers pay overblown prices to rescue predatory publishers? In my view, the services they offer at this time are far inferior to the support provided by Copernicus Publications (for example, the latter offers excellent language editing, a service that most major publishers have long abandoned.) No commercial publisher can have any valid excuse

for charging more than *Magnetic Resonance's* modest fee of 75-80 €/page, which is only a fraction of the current global cost (subscriptions and open access fees) of most publishers.

In the end, we have come to realize that the dispute about OA reflects the old divide between regulated and unbridled forms of capitalism: some believe that the economy must be kept under control by rules

and regulations, while others believe that it should be left to develop its dynamics freely. Until the emergence of monopolies leads to such a deep crisis that a reappraisal cannot be avoided. Needless to say on which side we stand.



## Footnotes and links

(1) The Ethics committee of the CNRS committee comprises a senior physicist (Michèle Leduc), an anthropologist (Antoinette Molinié), a chemist (Didier Gourier), an expert of immunology (Patrice Debré), an economist (Philippe Askenazy), an expert of artificial intelligence (Jean-Gabriel Ganascia), a biophysicist (Lucienne Letellier), a lawyer (Nathalie Nevejans), a

10    geophysicist (Catherine Jeandel), a philosopher (Frédérique Leichter-Flack), a mathematician (Jean Paul Delahaye), a theoretical solid-state physicist (Rémy Mosseri ), and a lawyer (Jean-Pierre Poussin ). The first two were *rapporteurs* of the "Opinion".

(2) https://peercommunityin.org

(3) https://comite-ethique.cnrs.fr/wp-content/uploads/2020/04/OPINION-2019-40.pdf.

15    (4) https://www.youtube.com/watch?v=OCfRV-8MYto. The event was chaired by Wim van Saarloos, with optimistic contributions by Johan Rooryck and Stan Gielen, with well-informed critical minority opinions by Joost Reek, Frits Rosendaal, Claartje Mulder, and Birgit Meyer.

20