# Peer review of "Open Access: Strengths, Weaknesses, Opportunities, and Threats."

_Magnetic Resonance, 2020_

## Referee Comment (RC1) · Gottfried Otting (Referee) · 6 Dec 2020

Open access is important, because it makes final peer-reviewed articles available to everybody from the day of publication. Open access has been shown to be associated with increases in citations, media attention, potential collaborators, job opportunities and funding opportunities (McKiernan et al., 2016).

Sadly, author pay charges (APC) associated with open access have led to a proliferation of predatory open access journals, which publish against payment without proper peer review and make authors suspicious about the quality of new open access journals.

At Magnetic Resonance, we are fortunate to have Copernicus Publications in Göttingen

(Germany) as the publisher. Copernicus Publications is a not-for-profit publisher who takes quality and ethical behaviour seriously. Articles are vetted before being put online and the final published version is carefully typeset to a high standard.

Furthermore, Magnetic Resonance practices a two-stage review system that has been successfully employed by many open access journals operated by Copernicus Publications over two decades. It fosters quality by transparency, making reviews as well as comments from the public available publicly on the forum called Magnetic Resonance Discussions. Students read these comments and the authors' responses with great interest!

Magnetic Resonance provides a platform for publications across a wide range of magnetic resonance subfields, including NMR and EPR spectroscopy, MRI and optical magnetic resonance phenomena, spanning physics, biology and biomedical applications. It aims to bring together researchers from a range of fields that have developed cultural differences over the years, such as the importance placed on the impact factor. At Magnetic Resonance, we feel that the value of an individual article is better measured by its citation numbers and downloads (both measured by Copernicus Publications) than by the impact factor of the journal. Nonetheless, we are certain that a quality journal, as Magnetic Resonance strives to be, will not have to hide once the impact factor becomes available after the first two years.

Geoffrey's editorial opinion piece correctly highlights the undue market power that established journals wield due to their name recognition and copyright privileges to past articles. For over 15 years, even a comprehensive initiative such as the PLOS endeavour has been unable to break the oligopoly of the established publishers. Recently, however, open access has been gathering momentum, with Plan S and DEAL being only two of the initiatives. For example, the Howard Hughes Medical Institute will expect its staff scientists to publish in journals moving to open access for all content (Brainard, 2020). Country-wide transformative agreements such as DEAL (Kupferschmidt, 2019) present a pathway to convert established journals into open access

publications which, without copyright privileges for past articles, will lose their grip on library budgets. Copernicus Publications has been competitive in this space for a long time, by championing agreements that transfer the cost of publication to library budgets rather than authors.

This reviewer does not share the author's despairing comments on different scientific cultures. Not all biology is butterfly collecting (as indeed acknowledged by the author) but also not all butterfly collecting is pointless – think of Charles Darwin's barnacle collection that bore the seed of the theory of evolution! Also, physics researchers may have embraced preprint servers like arXiv first, but the biological research community is coming on board, including the publication of referee reports (Guterman, 2020), similar to the Magnetic Resonance Discussions platform. Magnetic Resonance is thus well positioned for the future. Indeed, while its review process has been unfamiliar to most of us, it can be trusted to ultimately cater for a decent impact factor. How important will the impact factor be? Who knows, but on behalf of those who like to rank journals by their impact factor, I propose not to publish this editorial in Magnetic Resonance. The success of the journal should be measured by scientific research articles rather than opinion pieces.

Besides, the editorial will make more meaningful future reading in the context of other comments made on the discussion forum than as a standalone article.

References: Brainard, J.: News: HHMI mandates open access. Science, 370, 14–15, doi: 10.1126/science.370.6512.14, 2020.

Guterman, L.: People and Events: In biology publishing shakeup, eLife will require submissions to be posted as preprints. Science, 10.1126/science.abf9968, 2020.

McKiernan, E. C., Bourne, P. E., Brown, C. T., Buck, S., Kenall, A., Lin, J., McDougall, D., Nosek, B. A., and Ram, K.: Point of view: how open science helps researchers succeed. eLife, 5, e16800, doi: 10.7554/eLife.16800, 2016.

Kupferschmidt, K.: Deal reveals what scientists in Germany are paying for open access. Science, doi:10.1126/science.aax1064, 2019.

---

## Author Comment (AC1) · 7 Dec 2020

Gottfried Ottings is quite right to defend the noble art of butterfly collecting. Rutherford never said anything critical about these pretty insects either. The proper quote is: "All science is either physics or stamp collecting" (1925.) I strongly recommend http://rutherford.org.nz/msquotes.htm. If the readers find my editorial rude, let them be reassured that Rutherford was not always polite either when speaking about his colleagues: "He is like the Euclidian Point: he has position without magnitude" (c1912.)
* * *

---

## Referee Comment (RC2) · Daniella Goldfarb (Referee) · 8 Dec 2020

This editorial touches upon open access along with related topics, which are currently at the center of extensive discourse among scientists. How do we use bibliometrics to evaluate science? The power of the impact factor, the high APC costs of the so-called high impact factor journals, which adds to the high subscriptions costs and its consequences. It is an interesting read and it gives a valuable personal perspective along with relation to more general studies, like the one of the CNRS, which is quoted extensively. While most of the editorial is not strongly related to MR specifically , it does fit in MR discussion because it promotes discussion and thinking, particularly when MR represents the pretty face of open access. I think that this editorial should remain as part of the MR discussion and not as regular article because it does not report scientific

results and is an opinion manuscript. I do have a few specific comments: 1. I was very surprised at the 60,000 Euro expenses on OA during the first year of a personal grant. I knew OA can be very expensive, but that high ?  WOW 2.  The APC of Scientific Reports is not so high, a better example would be Nature Communications. 3.  The paragraph on p. 7 starting with "Magnetic resonance…" is written in plural. Not clear who "we" are . Particularly disturbing is the sentence "In our jaded assessment, their creativity is not always on a par with the founding fathers of magnetic resonance, save a few breakthroughs, such as the pursuit of minor conformations in proteins and nucleic acids." To whom "our" refer? This should be changed to "my" so it is clear that it is the personal view of the author.

---

## Short Comment (SC1) · 15 Dec 2020

Dear Prof. Bodenhausen, dear Geoffrey,

Let me start of by thanking you and everybody involved for launching Magnetic Resonance (MR). I think it is a very important initiative for the magnetic resonance community, and I am very happy that I could contribute during its first year.

One of the nice things about MR is that even Editorials can be discussed.

Unfortunately, I am not entirely sure what the main point of it is. The title includes "Strengths, Weaknesses, Opportunities, and Threats" (SWOT), which is a common part of strategic planning and evaluations, for example in companies, but also Universities. On first sight, I hoped that you would do such an analysis for MR, but I do not think

this is really the case. Then I recognized that the title and abstract actually suggest that a SWOT analysis would be performed for "Open Access" in general. This would also be valuable (although it would not be in the usual sense of SEOT), but again I think the editorial is not substantial enough for this. It does address some points, but sometimes rather superficially. At times it reads more like a "book review" of the mentioned 'opinion' of the Ethics Committee of the French CNRS.

To the points regarding MR in particular:

- It would be nice to give some numbers to the general feelings. i.e. you mention that "Most papers received so far come from European laboratories". I could look it up myself and count, but why not give numbers?

- As mentioned by Prof. Daniella Goldfarb, it is unclear who "we" is. It suggests that it is the Editorial Board, but this is apparently not the case.

- You attribute the fact that there are less Biology papers in MR (are there? Numbers?) to a "unbridled passion for impact factors" in the biomolecular community. First, this sounds like your feeling. It might be true, but can you put numbers to this? Did anybody look at this in more detail? Second, it sounds like a cheap shot. If someone works on a biological problem, uses NMR, maybe cryoEM on top, and uses biochemical studies as well, why would that scientist hand in an article to MR? Was this the case in, e.g. the Journal of Magnetic Resonance (JMR)? To me it simply sounds like such studies are "out of scope", which is not problematic at all. At the same time, you discredit such studies and assess that "their creativity is not always on a par with the founding fathers of magnetic resonance" (I think that is a rather high bar...). I am not sure what you want to say. Do you want more biology papers in MR, or not?

- In general, the tone is sometimes rather irritating. I assume this was on purpose, and I do not think it is inappropriate for an Editorial, but I am not convinced that you gain authors for MR like this.

- Unfortunately, there were no words regarding the development of MR. Do you or Copernicus Publications intend to implement some changes? Did everything in the editorial processes run smoothly this year or were there problems? Did anybody ask for waivers of the publication fees? Was this granted (of course anonymous information would suffice here)? i.e. does MR contribute to less inequality in the publication process?

Regards, Nino Wili

---

## Author Comment (AC2) · 20 Dec 2020

In her review of my "editorial", Daniella Goldfarb expressed her wish to know more about the outrageous hybrid charges of € 60 000 that I claimed to have paid in 2014. Assuming that her curiosity is shared by others, I have done some homework on a confined rainy Sunday. I have broken down our extravagant payments according to journal. In many instances, I found the original invoices of 2014-15, including no less than four reminders in a case that I challenged in vain. For other papers, I have taken the current APC rates for hybrid open access (December 2020) as indicated on the websites of Wiley, Elsevier, PNAS, RSC, ACS, etc. After conversion into Euros at current rates, and after adding 20% VAT applicable in France, I could reconstruct the following expenses for the 16 papers of which I was a co-author in 2014. Of course,

this is a personal and rather arbitrary compilation. Many readers and authors who have contributed to 'Magnetic Resonance' will recognize familiar titles:

Dalton Transac. 2112 € ; Chem. Phys. Lett. 3156 € ; J. Phys. Chem. B 5100 € ; Chem. Eur. J. 3000 € ; PNAS 2550 € ; Angew. Chem. 3500 € ; ChemMedChem 2500 € ; J. Magn. Reson. 3420 € ; Chem. Phys. Lett. 2279 € ; J. Phys. Chem. Lett. 5100 € ; Magn. Reson. Chem. 3348 € ; Chem. Eur. J. 3000 € ; Angew. Chem. 3500 € ; ChemPhysChem 2500 € ; Phys. Chem. Chem. Phys. 2112 € ; RSC Advances 0 € .

This adds up to 47 177 € for the year 2014. I must apologize to the readers of 'Magnetic Resonance': the estimate of 60 000 € given in my editorial was inflated by 27%, which I'm tempted to attribute in part to the sense of hyperbole of our charming accountant. The reason I chose to pay these outrageous hybrid Open Access charges was the following rule: "For ERC Frontier Research Grants funded under FP7 for which the Grant Agreement contains Special Clause 39 ERC it is mandatory to enable open access." To be truthful, I could never determine whether Special Clause 39 applied to our work or not, my ERC grant being merely "advanced", which is presumably less critical than "at the frontier". In the subsequent 5 years of my ERC grant (2015-2019), I decided to ignore this clause, and stopped paying article page charges (APCs). So far, the ERC has not complained.

Of the 15 hybrid ("gold OA") papers that we published in 2014, the total number of pages of is 90 (not counting the OA paper in RSC Advances), for which we paid 524 € per page or 3145 € per paper. By comparison, 'Magnetic Resonance' publishes papers for a mere 80 € per page if the paper is formatted in Word, reduced to 75 € per page if submitted in LaTex. Furthermore, these article page charges can be waived, but we have only received two requests for the first 35 submissions.

While retrieving mails from early days, I found some entertaining messages. To the 'Journal of Magnetic Resonance', I wrote in September 2012: "I enclose a copy of an

invoice for € 1185,88 for some proofs that were handled in an exceptionally sloppy manner by your office. Even the third set of proofs contained several errors. I estimate that the additional expenses incurred by my co-workers and myself, due to your practice of employing cheap and inexperienced co-workers, exceed the sum of € 1500. Shall I send you an invoice on my EPFL letter heading, or shall we agree to cancel the two invoices against each other ?". To my surprise, Elsevier decided to drop its invoice.

More recently, in 2019, I wrote to 'Progress in NMR Spectroscopy': "In recent months, Elsevier has turned out to be incapable of 'rolling out' decent products. A recent paper submitted (on Geoffrey's invitation) by Luchinat et al. to PNMRS required over 659 corrections for a paper of only 26 pages, and no less than 4 consecutive sets of proofs. Another paper submitted to PNMRS by Andrew Pell on Gareth Morris' invitation contained no less than 1,734 errors in the first proof. When we complained, [Elsevier] replied that these errors must be ascribed to the "tools" that Elsevier uses." There is no indication that Elsevier is willing to improve these tools.So far, I have not heard of any similar complaints by authors who have contributed to MR.

Finally, I found a heart-warming announcement on the website of the RSC "In their first year, articles published OA with us are downloaded 97% more often than non-OA work." This is my take-home message for authors who hesitate to choose 'Magnetic Resonance'!

---

## Author Comment (AC3) · 20 Dec 2020

In reply to Nino Willi's comments: when we started writing our 'editorial', the initial idea was merely to attract attention to the 'opinion' of the Ethics Committee of the French CNRS, in the hope that our readers would immerse themselves in the subtleties and intrinsic contradictions of Open Access (OA). Our use of the pronoun 'we' (also mentioned in Daniela Goldfarb's review) is known as 'pluralis majestatis', which prescribes the use of a plural pronoun when referring to a single person, commonly employed by persons of high office. This does not mean that we aspire to be recognized as a monarch. My marvelous, though delightfully old-fashioned high-school teachers in Geneva taught me to avoid the use of the singular 'I' when writing papers. For the sake of clarity, I will stray from their prescriptions in the next few paragraphs.

[Figure]

The idea of writing an extensive SWOT analysis of 'Magnetic Resonance', let alone of Open Access, would run against my principles. In fact, I used the expression SWOT ironically, for I believe that granting agencies and other incarnations of management who insist on compiling unwieldly SWOT tables are invariably spoon-fed with the same evasive excuses. SWOT is one of those inherently inadequate 'management tools' that can only lead us astray from our lofty objectives.

Clearly, authors who are serious about biology should publish in biology journals. As Nino Willi must realize all too well, it is pointless to twist the arms of bio-authors in the hope that they submit their work to 'Magnetic Resonance'. Indeed, their papers may well be out of scope, as Nino rightly presumes. As for the admittedly inimical comment about the "unbridled passion for impact factors" that I attributed to some of the biomolecular community, this is indeed a strictly personal perception, supported by my judgment of the dubious role of Nature, Cell, PNAS, etc., and the ruthless ambition of some prominent bio-NMR colleagues. Like the rebellious Prometheus who stole the sacred fire of Olympus against Zeus' wishes, bio-NMR specialists are forever condemned to seek financial support for their expensive hobbies. Not surprisingly, bio-NMR pundits have become remarkably proficient in attracting both recognition and plentiful resources, not least by publishing in highly visible journals. Their fate of perpetual money-seekers is surely less cruel than Prometheus' who was ruthlessly punished for his gift of fire to mankind by having his liver picked by Zeus' eagle day after day through an open wound in this belly. Perhaps the equivalences

specialists of bio-NMR = Prometheus

magnetic resonance = sacred fire

biologists = mankind

reveal my simplistic taste for structuralism-for-beginners. Besides, none of the editors of 'Magnetic Resonance' claim Zeus' authority to inflict punishment.

[Figure]

The condescending tone that Nino Wille found 'irritating' was thus intended. Mind you, I have high regard for some people working in bio-NMR, not least in my own lab in Paris, where some team members occasionally remind me, ever so kindly, that the triumphant days of pure methodology may well belong to the past. Yet I cannot deny that I have been disappointed by the lack of creativity of many bio-lectures that I followed via Zoom in the last 10 months. There are only so many structures of proteins and their complexes that I can watch before their soporific effect sets in. In many cases, though not in all, I have grave doubts whether their structure really allows one to understand their function, which is the underlying hypothesis of much of this work. In terms of entomology, describing the anatomy of an ant doesn't necessarily allow one to understand how an ant colony functions.

Perhaps my irritability is fueled in part by the fact that I myself tried hard to contribute to bio-NMR. I'm not thinking about trivial inventions such as HSQC, but about relaxation-allowed coherence transfer (RACT), cross-relaxation with suppression of spin diffusion (QUIET NOESY), curious consequences of cross-correlation (CCCC), tensor operators of rank l and coherence order p (Tlp) as tools to expand density operators for methyl groups, etc. By and large, these attempts to contribute to bio-NMR (I dare say: heroic attempts, from my perspective) have sunk into oblivion without a leaving bubble. Most bio-NMR pundits are trying too hard to compete against real biologists to educate themselves on spectroscopy.

Intrigued by Nino Willi's question, I checked that NONE of the corresponding authors of the first 25 papers published in 'Magnetic Resonance' are working in the USA. So far, 'Magnetic Resonance' has enjoyed support from Australia (1 paper), New Zealand (1), the Netherlands (1), the United Kingdom (1), Austria (1), Sweden (1), Israel (1), Russia (1), France (2), Switzerland (5), and Germany (9). The last number can be attributed in part to Elsevier's unwillingness to strike an agreement with the German DEAL. In my opinion, none of the first 25 papers can be ranked as 'biological', although some describe applications to biological samples.

'Magnetic Resonance' would be glad to receive, publicly review and publicly comment papers from bio-NMR and non-bio-NMR colleagues alike, in the Americas, Africa, and Asia, and publish them for a mere 80 € per page if the paper is formatted in Word, reduced to 75 € per page if submitted in LaTex, and further reduced to naught if a waiver is requested and granted. Encouraged by the quality of the first 35-odd submitted papers, we shall work with Copernicus Publications to implement some improvements of the web pages, to make sure that the authors are not confused by the unusual aspects of publicly accessible reviews, and to make the editorial processes run even more smoothly.